# Simple Bayesian reconstruction and forecasting of stream water temperature for ecologists—A tool using air temperature, optionally flow, in a time series decomposition approach

**Guillaume Bal** [1]*, **Elvira de Eyto** [2]

**1** PatriNat (OFB, MNHN), Brunoy, France, **2** Fisheries Ecosystems Advisory Services, Marine Institute, Furnace, Newport, Ireland

* guillaume.bal.pro@gmail.com

**Data Availability Statement:** All the data used in this study for the Rough are published and accessible here: http://data.marine.ie/geonetwork/

## Abstract

Mitigating the impacts of global warming on wildlife entails four practical steps. First, we need to study how processes of interest vary with temperature. Second, we need to build good temperature scenarios. Third, processes can be forecast accordingly. Only then can we perform the fourth step, testing mitigating measures. While having good temperature data is essential, this is not straightforward for stream ecologists and managers. Water temperature (WT) data are often short and incomplete and future projections are currently not routinely available. There is a need for generic models which address this data gap with good resolution and current models are partly lacking. Here, we expand a previously published hierarchical Bayesian model that was driven by air temperature (AT) and flow (Q) as a second covariate. The new model can hindcast and forecast WT time series at a daily time step. It also allows a better appraisal of real uncertainties in the warming of water temperatures in rivers compared to the previous version, stemming from its hybrid structure between time series decomposition and regression. This model decomposes all-time series using seasonal sinusoidal periodic signals and time varying means and amplitudes. It then links the contrasted frequency signals of WT (daily and six month) through regressions to that of AT and optionally Q for better resolution. We apply this model to two contrasting case study rivers. For one case study, AT only is available as a covariate. This expanded model further improves the already good fitting and predictive capabilities of its earlier version while additionally highlighting warming uncertainties. The code is available online and can easily be run for other temperate rivers.

## Introduction

Global warming is impacting ecological communities and ecosystems worldwide [1–3]. Temperature primarily impacts the physiology of organisms [4], leading to changes in

srv/eng/catalog.search#/metadata/ie.marine.data:
dataset.5122. In case of the Scorff river, data are
accessible free of charge as well. Water
temperature are accessible through a DOI too. Flow
data are hosted onto the French hydrology services
database and are downloadable at https://hydro.
eaufrance.fr/stationhydro/J510221001/series. In
case of the air temperature time series, a data
sharing agreement with the French Weather
Services is necessary for gratuitous access by
scientists. Please go to https://donneespubliques.
meteofrance.fr/?fond=dossier&id_dossier=1 for
details.

**Funding:** The author(s) received no specific
funding for this work.

**Competing interests:** The authors have declared
that no competing interests exist.

individual life history traits [5–7]. Populations range [8, 9], phenology [10] and dynamics are also impacted [11], driving disturbances in food webs and overall ecosystem change [12–14]. Managing these ongoing and future changes is a major challenge for stakeholders.

Understanding and hence mitigating climate change impacts on biological communities involves four steps. First, we need to quantify the interactions between the process of interest and temperature. Second, we need to predict temperature under future greenhouse gas (GHG) scenarios. Third, we need to forecast our process under these future climate scenarios. Then, as the fourth step, we can work towards mitigating impacts, preferentially through an adaptive management framework [15, 16]. Robust ecological and temperature data are essential corner-stones of this process.

Managing freshwater species conservation in the face of climate change is particularly chal-lenging. First, such species are highly susceptible to climate change as they are mostly ectother-mic and thus sensitive to temperature [17–19]. Shifts in spatial distribution are also constrained by the river network and habitat fragmentation [20]. Second, water temperature time series are limited compared to those of air temperature. Indeed, water temperature time series are often quite short and prone to missing data, and unlike air temperature, are not gen-erally included as outputs of climate change models. Generic models to complete and hind-cast/forecast water temperature based on covariates such air temperature and flow are needed by ecologists and managers.

Different types of stream water temperature models exist but they present caveats to the ecologist [21]. Process based models, e.g. energy budget models [22–24] are realistic but complex. They require a lot of environmental and geological data, and are also loca-tion specific. This makes them of limited use for long term and large-scale forecasting. Statistical models on the other hand are less data demanding and rather simple [25]. Examples of this model type include regular time series models (AR, ARIMA), periodic autoregressive models, K nearest neighbors methods and neural network approaches [26, 27]. These methods perform well for short term forecasting and filling short gaps. On the contrary, they perform poorly for long term predictions [25]. The last category usually encountered is regressions models [28–30].While they may seem adequate, they can lead to significant biases in more long-term forecasting. Indeed, they do not disentangle long term trends, seasonality and short-term variations. An extensive review of stream water temperature models can be found in [31] and allows for a better appraisal of the limita-tions exposed.

As a workaround, [21] proposed a Bayesian hierarchical approach which was a hybrid between time series decomposition and regression. This approach required water tempera-ture together with air temperature and flow time series as covariates. It separated out long term, seasonal and short terms components of the time series and linked these components through correlation. This method has several advantages over regression, in that it outper-formed correlation in both fit and forecasting capabilities, and appeared to be unbiased for long term forecasting. This hybrid approach also highlighted the true, and high uncertainty, in future river temperature warming. However, this approach lacked fine-scale modelling of deviations (i.e. daily) around long term and seasonal components, and required addi-tional refinement. Specifically, short-term deviations were modeled using a first order auto-regressive (AR1) process which limits fine hindcasting/forecasting abilities, and the model worked on 5-day average temperature which is rather coarse. In this paper, we improve on the previous model by i) working on daily temperature and ii) modeling daily deviations as depending on that of air temperature (AT) and flow (Q). Lastly, we offer the possibility of running this model without water flow data, as these are often not collected routinely.

These new features, coupled with openly accessible code should be helpful to ecologists and stakeholders.

In this paper, we describe the fit and forecasting performances of this extended model and compare it to that of [21] on two rivers with contrasting bio-geographical conditions and sizes. The Rough River is a small Irish spate river while the Scorff river is of medium size and located in France. For the Rough river, air temperature is the only covariate. As in [21], we use a fully hierarchical Bayesian framework. This provides a probabilistic rationale to quantify uncertainty in inferences and forecasting which further ease out ecological analysis and risk management [32].

## Material and method

The new model version presented here is developed with reference to that of [21]. For this reason, we refer to the model in the previous publication as $M_1$ (for model 1). Our newest version described thereafter is called $M_2$.

## Model $M_2$ structure

Model $M_2$ has a fully Bayesian hierarchical structures and produces estimates of water temperature (WT) based on air temperature (AT) and discharge (Q). This is achieved using three fully integrated modules. Module 1 desegregates WT, AT and Q time series into long term variations, seasonal fluctuations and short-term variations. Module 2 links WT components to that of the AT and Q (also referred to as covariates). Module 3 then provides estimates of all unknown WT based on modules 1 and 2. Estimated WT can range anywhere from a few missing values up to several years in case of hindcasting or forecasting. The difference between model $M_1$ [21] and $M_2$ lies in module 2 and also in the way daily variation in WT are modeled. In case of model $M_1$, daily variation was assumed to follow an autoregressive process. Here we expand module 2 to link these to those of the covariates.

Module 1 of the model $M_2$ desegregates WT and covariates time series ($X_t$ in the following equation) as follow:

$$X_{y,t} = \alpha_y + \beta_y \times sin\left(\frac{2\pi}{n}(t - t_0)\right) + \epsilon_t \tag{1}$$

$\alpha_y$ and $\beta_y$ are the mean and amplitude of a time window. $n$ is the number of time steps per year. In this article $n$ is equal to the number of days within a year (365 or 366) while we set the time window $y$ to 6 months as it offers a good tradeoff (see [21]). $t_0$ sets the seasonal signal on the year. For AT and Q, we modeled $\epsilon_t$ using first order autoregressive (AR1) processes centered on 0 (with autocorrelation coefficient $\rho$ and standard deviation $\sigma$). [21] modeled WT daily deviations the exact same way for model $M_1$. This is where this new model differ and we detail this as part of the second module below.

Module 2 links WT time series to that AT and Q (if available). It comprises two sub-modules (2a and 2b). Sub-module 2a corresponds to the entirety of module 2 from [21]. In this first sub-module, we link parameter $\alpha_y$ and $\beta_y$ of the WT time series to that of the covariates time series. To do so, we first reparametrized the sine signal within the time series using $max_y = \alpha_y + \beta_y$ and $min_y = \alpha_y - \beta_y$. We then model WT maxima $max_y^{WT}$ and minima $min_y^{WT}$ as depending

upon those of the covariates. Corresponding equations are:

$$max_y^{WT} \sim Normal\left(\mu_{max_y^{WT}}, \sigma^2_{max^{WT}}\right)$$

$$with\ \mu_{max_y^{WT}} = \theta_0 + \theta_1 \times max_y^{AT} + \theta_2 \times min_y^{Q}$$

$$min_y^{WT} \sim Normal\left(\mu_{min_y^{WT}}, \sigma^2_{min^{WT}}\right) \tag{2}$$

$$with\ \mu_{min_y^{WT}} = \theta_0 + \theta'_1 \times min_y^{AT} + \theta'_2 \times max_y^{Q}$$

We kept this parametrization from [21] because of its conceptual sense. Let's assume that $\theta_1$ is positive, $\theta_2$ is negative and both $\theta'_1$ and $\theta'_2$ are positive. This corresponds to assuming that i) warm AT leads to warm WT, ii) high Q means warms WT in winter and a cooling effect in summer.

In addition, module 2b now links the short-term variations of WT to that of the covariates. We parametrized this addition to the original model $M_1$ as follows:

$$\epsilon_{y,t}^{WT} = \delta \times \epsilon_{y,t}^{AT} + \gamma \times \epsilon_{y,t}^{Q} \times sin\left(\frac{2\pi}{n}(t - t_0 + n/2)\right) + \zeta_t^{WT} \tag{3}$$

WT short term deviations thus are directly proportional to those of AT, although usually buffered. For Q, we included a sinusoidal component. This allows the impact of discharge deviations to change along the year. It shifts from positive in winter to negative in summer with a smooth transition along the year. We assume it as strictly antiphasing the WT seasonal signal, hence the inclusion of $n\backslash 2$. Because we use the log discharge, small changes in summer flow have an impact more comparable to that of bigger changes in winter. We modeled the remaining errors ($\zeta_t^{WT}$) using a white noise process.

Lastly, module 3 corresponds to the estimate of unknown temperatures. As our model is fully hierarchical, this module depends upon and integrates with modules 1 and 2. Consequently, the model propagates all uncertainties when estimating unknown WT based on covariates. In Bayesian software such BUGS, JAGS, STAN or NIMBLE ([33] and references therein), this is performed simultaneously to the fit. AT and Q time series just have to include hindcasting and/or forecasting parts if any. Module 1 also decomposes these 'extra' data while module 2 produces the WT estimates. This process is often referred to as deriving posterior predictive distributions.

## Model application

To assess the performances of the model ($M_2$), we compared it to model ($M_1$) on two study sites with contrasting sizes and environmental conditions and tested both fit and forecasting capacities.

**Assessing models' performances.** We checked the consistency of both models $M_1$ and $M_2$ a posteriori with the data using the $\chi^2$ discrepancy statistics (Gelman et al. 2015). In particular, this posterior check allows to assess whether replicated data are similar to the original data. Its formula is:

$$\chi^2(WT, \theta) = \sum_y \sum_t \frac{\left(WT_{y,t} - E\left(WT_{y,t}|\theta\right)\right)^2}{Var\left(WT_{y,t}|\theta\right)} \tag{4}$$

where $E(WT_{y,t}|\theta)$ and $Var(WT_{y,t}|\theta)$ are the expected mean and variance of WT conditionally upon all parameters ($\theta$). For each set of parameters drawn from the joint posterior distribution, we computed replicated and observed WT $\chi^2$ values based on the equation. Then we computed a Bayesian p-value giving the probability that $\chi^2(WT_{rep}|\theta) > \chi^2(WT_{obs}|\theta)$. Bayesian p-values close to 0.5 suggest congruency between the model and the data. Very high or very low (about 0.95 or 0.05) values provide serious inconsistency warnings.

Secondly, we also compared the fitting performances of models $M_1$ and $M_2$ using the Deviance Information Criterion (DIC, [34]). DIC allows the comparison of goodness of fit while penalizing complexity in a way very similar to that of the Akaike Information Criterion. The smaller the DIC the better. A five points reduction is usually considered as indicative of a significant improvement.

Lastly, we checked the predictive performances of the models. For this we used cross a validation approach. Specifically, we used 2/3 of historical data as training set and forecast the last third. We used the Root Mean Square Error (RMSE) of the estimates to compare performances.

**Bayesian computations details.**   We implemented models $M_1$ and $M_2$ in JAGS [35] through R [36, 37]. This software approximates the parameters posterior distributions using Monte Carlo Markov Chains (MCMC) algorithms. Specifically, we ran three MCMC in parallel for each model fit. We kept 10000 draws for each one after both a thinning of 25 and an initial burning period of 10000 draws. All diagnostics suggested the MCMC converged successfully. Priors used were all uninformative relative to the data (see S1 Table in S1 File).

The code developed together with one of the data-sets are available on GitHub (https://github.com/GuillaumeBal/2023.bayes.stream.temperature). Readers can change the input data file with their own and run the model. Doing so does not require any in depth knowledge of both R and JAGS.

**Study sites.**   WT, AT and Q time series come from the Rough (Mayo, Ireland) and Scorff (Brittany, France) rivers (Fig 1). Both rivers support long-term environmental monitoring and fisheries related research [5, 43, 44]. They contrast in size, location, as well as available data (Table 1). In particular, reliable Q time series are lacking on the Rough river. This allowed for testing the performances of our approach with only one covariate.

The Scorff River flows into the Atlantic Ocean (Fig 1). INRA staff members measure daily WT at the Moulin des Princes station with Tidbit data loggers (precision of 0.2˚C, [45]). Daily AT come from the Lorient (LannBihoueé airport meteorological station operated by the French weather services (https://meteofrance.com/), 9 km away from the water temperature measurement station. Q records come from 8 km upstream of the Moulin des Princes station where a flow measuring station is operated by the French hydrological services (https://hydro.eaufrance.fr). The three time series are 13 years long (1995 to 2007) and 9.6% data are missing on average. This river is part of a larger research observatory where diadromous fish population are extensively monitored and studied (ORE DiaPFC).

The Rough River (also called the Srahrevagh River) is a tributary of the Srahmore River in the Burrishoole Catchment. The Srahmore flows into Lough Feeagh which is connected to the saline coastal lagoon Lough Furnace. The entire system discharges into Clew Bay on the Atlantic Ocean (1. Data cover the period 2002–2016. Daily WT was measured using a StowAway TidbiT temperature data logger from Onset (TB132-05+37). Air temperature data was recorded at the Newport (Furnace) manual weather station (jointly run by Met Éireann and the Marine Institute, station 833). Maximum and minimum temperatures are recorded using mercury thermometers, and average daily temperature is taken as the average of these two

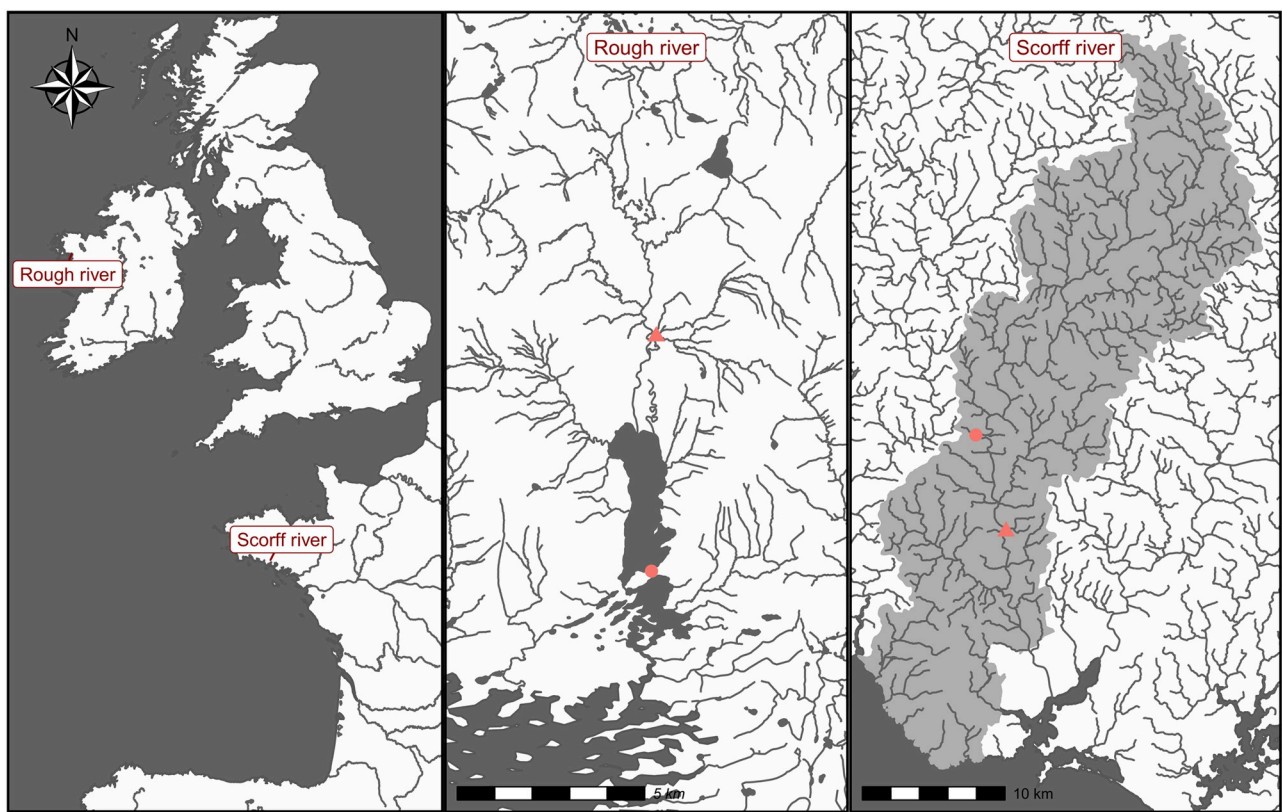

**Fig 1. Map of the application sites.** Triangles correspond to water temperature and flow measurement stations while points indicate weather monitoring stations. The map was drawn based on CC-By 4.0 compatible shapefiles [38–40] combined together using some R packages [36, 41, 42].

readings. Data were downloaded from www.met.ie, licensed under a Creative Commons Attribution 4.0 International (CC-By 4.0) License.

Some more details about the catchments are provided in Table 1. Raw time series are shown within the supporting information file (S1 Fig in S1 File).

**Forecasting under climate change.** We used full joint posteriors from model $M_1$ and $M_2$ to project WT temperature from AT warming projections. In particular, we picked a range of AT warming of 0 to 5°C according to the latest IPCC estimates [46], excluding the most extreme warming scenario. We did this for several reasons. First, [21] revealed WT warming should be lower than that of AT and quite uncertain. We wanted to check whether this result was robust to the updated time step and model structure. We also wanted to see whether two rather contrasted rivers would differ in their response.

## Results

The results section is composed of four parts. First, we show differences in fitted series of $\alpha$ and $\beta$ with both models. Then, we focus on quantitative results showing how the updated approach of modelling the water temperature residuals as depending upon those of air temperature and flow outperformed the version with a simple AR1 process, both for forecasting and fitting. Third, we look at parameters of model $M_2$ and highlight how those shared with the first model evolved to better understand model differences. Lastly, we showcase forecasting and missing value estimates from model $M_2$.

**Table 1. Application sites and data sets description.**

| River | Rough | Scorff |
|---|---|---|
| Location | Mayo, Ireland | Brittany, France |
| Mouth (Latitude & Longitude) | 53˚97'N, -9˚57'W | 47˚28'N, 3˚23' W |
| Drainage area (km$^2$) | 4.60 | 480 |
| River Length (km) | 12.3 | 75 |
| Estuary length (km) | NA | 15 |
| Source altitude (m above sea level) | 370 | 270 |
| Geology (predominant) | Quartzite & schist | Granite & schist |
| % agricultural | NA | 60 |
| % woodland | NA | 30 |
| % peat bog | 3.2 | NA |
| % forestry | 32.2 | NA |
| % natural grassland | 7.9 | NA |
| Climate | Temperate oceanic | Mild oceanic |
| Precipitations(mm) | ~1560 | ~1000 |
| Water temperature period | 2002–2016 | 1995–2007 |
| Water temperature mean (˚C) | 9.91 | 12.88 |
| Water temperature missing data (%) | 0.82 | 26.98 |
| Air temperature period | 1960–2016 | 1995–2007 |
| Air temperature mean (˚C) | 10.25 | 12.39 |
| Air temperature missing data (%) | 0.21 | 0.63 |
| Water discharge period | NA | 1995–2007 |
| Water discharge mean ($m^3 . s^{-1}$) | NA | 4.95 |
| Water discharge missing data (%) | NA | 1.05 |

## Seasonal variation in the time series

Comparing time series of fitted $\alpha$ and $\beta$ between models $M_1$ and $M_2$ provides basic insights into differences in their fit and behavior. The ranges of values are greater for model $M_2$ than for model $M_1$ whatever the parameter series and river considered (Fig 2). This results in statistically significant changes in the variance of median values. For instance, the variance of the median estimates of the $\alpha_{WT}$ posterior series on the Scorff river was 0.47 for model $M_2$ versus only 0.17 with model $M_2$ (p.value = 0.01, Fischer test). While series ranges differed, mean values of median estimates for all series were equivalent. The same was observed for average 95 and 50% credible intervals.

In case of AT and Q, $\alpha$ and $\beta$ time series derived using both models were quite similar (S2, S3 Figs within the S1 File). Mean values, variances and 95 and 50% credible intervals of the time series deviated from each other only fractionally. Only a few posteriors, such as the last couple for the Rough river $\alpha_{AT}$, were obviously different.

## Fit and predictive performances comparison

Overall, results suggest better predictive performances for model $M_2$ on both rivers. In addition, fit statistics that indicate good predictive performances were more apparent for model $M_2$. $\chi^2$ discrepancies and associated p-values were better on both rivers with model $M_2$ but p-values obtained with model $M_1$ did not depart enough from 0.5 to be indicative of an inconsistency between this model and the data (Table 2).

The only exception to the general superior performances of model $M_2$ were DIC values (Table 2) for the Scorff River. Indeed, differences of several hundred of points in deviance in

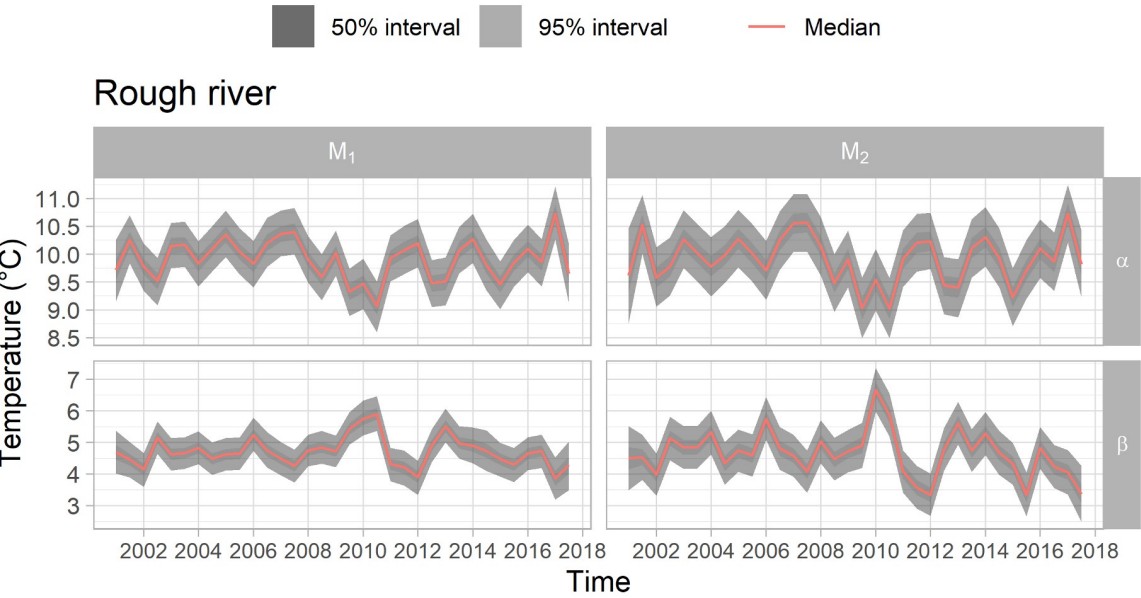

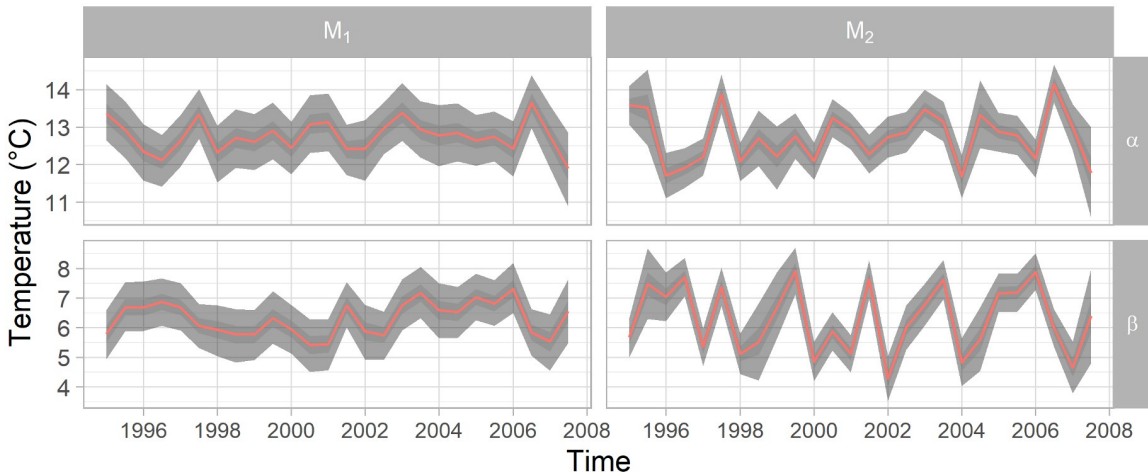

**Fig 2. WT $\alpha$ and $\beta$ parameters posteriors for model fit.** In case of model $M_2$ WT residuals are linked to those of AT and Q.

**Table 2. Fitting and forecasting performances summary statistics.**

| River | Model | RMSE | $\chi^2$ p-value | Deviance | pD | DIC |
|---|---|---|---|---|---|---|
| Rough | $M_2$ | 1.34 | 0.50 | 12635 | 289 | 12925 |
| Rough | $M_1$ | 2.28 | 0.43 | 15585 | 31.1 | 15616 |
| Scorff | $M_2$ | 1.86 | 0.50 | 10763 | 46.8 | 10810 |
| Scorff | $M_1$ | 2.87 | 0.40 | 7335 | 29.2 | 7364 |

Deviance: deviance posterior mean; pD: measure of the model complexity (estimated number of parameters); DIC: Deviance Information Criterion; $\chi^2$ p-value: p-value for the posterior checking tests; RMSE: root mean square errors used to quantify the predictive performance.

favor of model $M_1$ were highlighted by the analysis on that river. The same was consequently observed for the DIC value. This better fit performance of model $M_1$ on the Scorff river resulted from its highest fit flexibility when a lot of data are missing that is linked to modeling the residuals with an AR1 process. We also noted that the lack of covariate data tends to increase pD values in case of model $M_2$ as observed for the Rough river.

Discrepancies between 6-month averages of replicated data and observed water temperature were clearly in favor of the more complex model $M_2$ (Fig 3). Indeed, related posterior distribution for $M_2$ appeared both better centered on zero but also exhibited a much narrower credible interval. For the Scorff river, the average 95% interval dropped from 1.63 to 0.48°C. For the Rough River, the corresponding values decreased from 0.99 to 0.29°C. We can also see that the distributions for the second half of years 2009 and 2010 on the Rough River are far

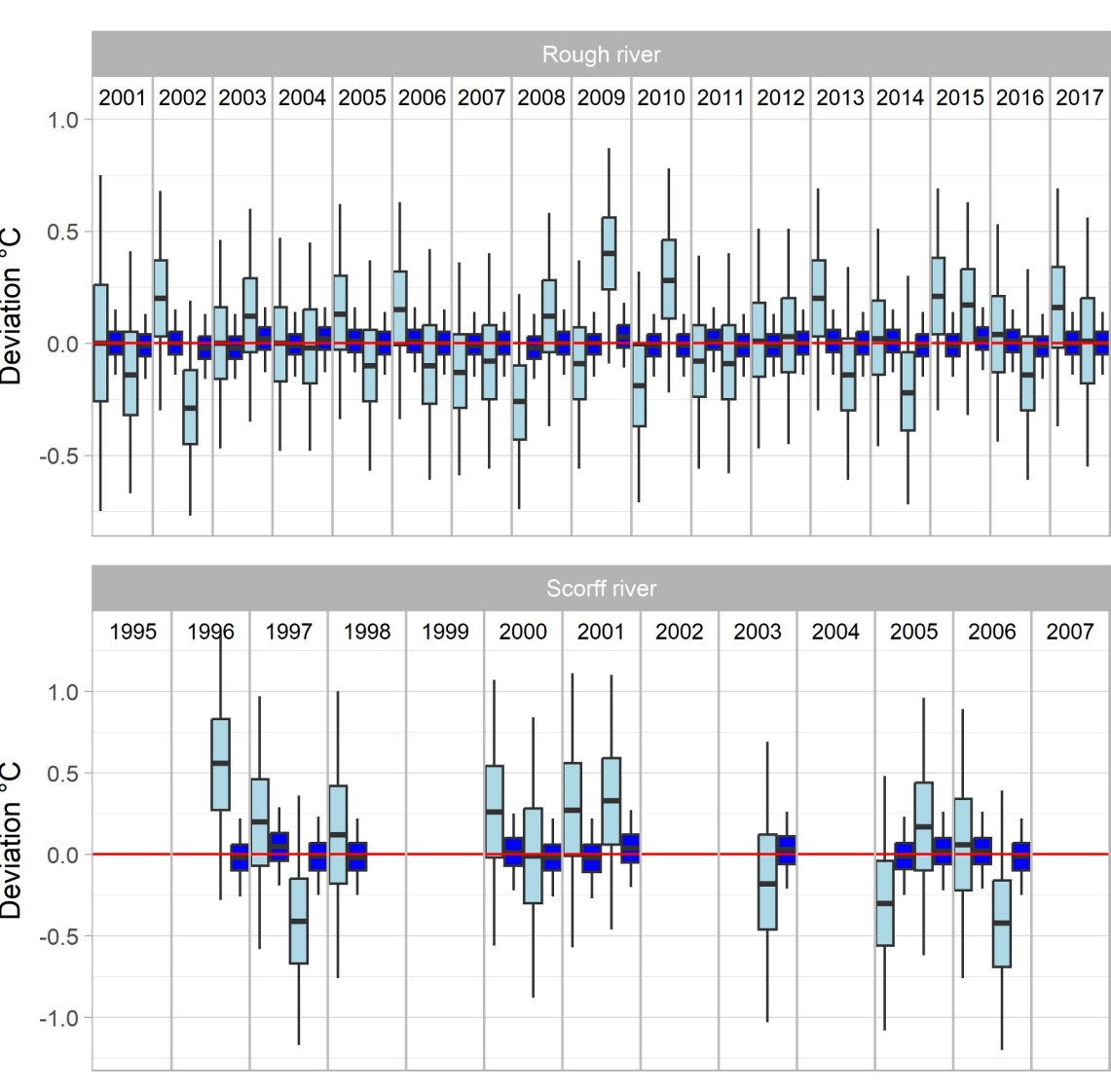

**Fig 3. Boxplot of discrepancies between observed and fitted mean six months temperature.** Model $M_1$ features an AR1 modeling of WT residuals. In case of model $M_2$, WT residuals are linked to those of AT and optionally Q.

from being centered on zero in case of the old model $M_1$ (Fig 3). We noticed the same problem with the second half of years 1996 and 2006 on the Scorff river.

RMSE values clearly highlighted the better forecasting performances of model $M_2$. Indeed, incorporating the link between residuals when modeling water temperature allowed for a reduction in RMSE of 41 and 35% for the Rough and Scorff River respectively (Table 2). The forecasting capabilities of model $M_2$ are further highlighted on Fig 4. Here, we selected a small portion of historical water temperature data from each river, containing both low and high daily variations. On both rivers, observed and predicted daily variations follow each other closely. High frequency changes are well captured by model $M_2$, as can be seen for early summer (days 75–100) for both rivers (Fig 4). The average 50% credible interval around predictions is about 0.75˚C for both rivers, while the 95% credible interval spans slightly more than 4.5˚C (Fig 4).

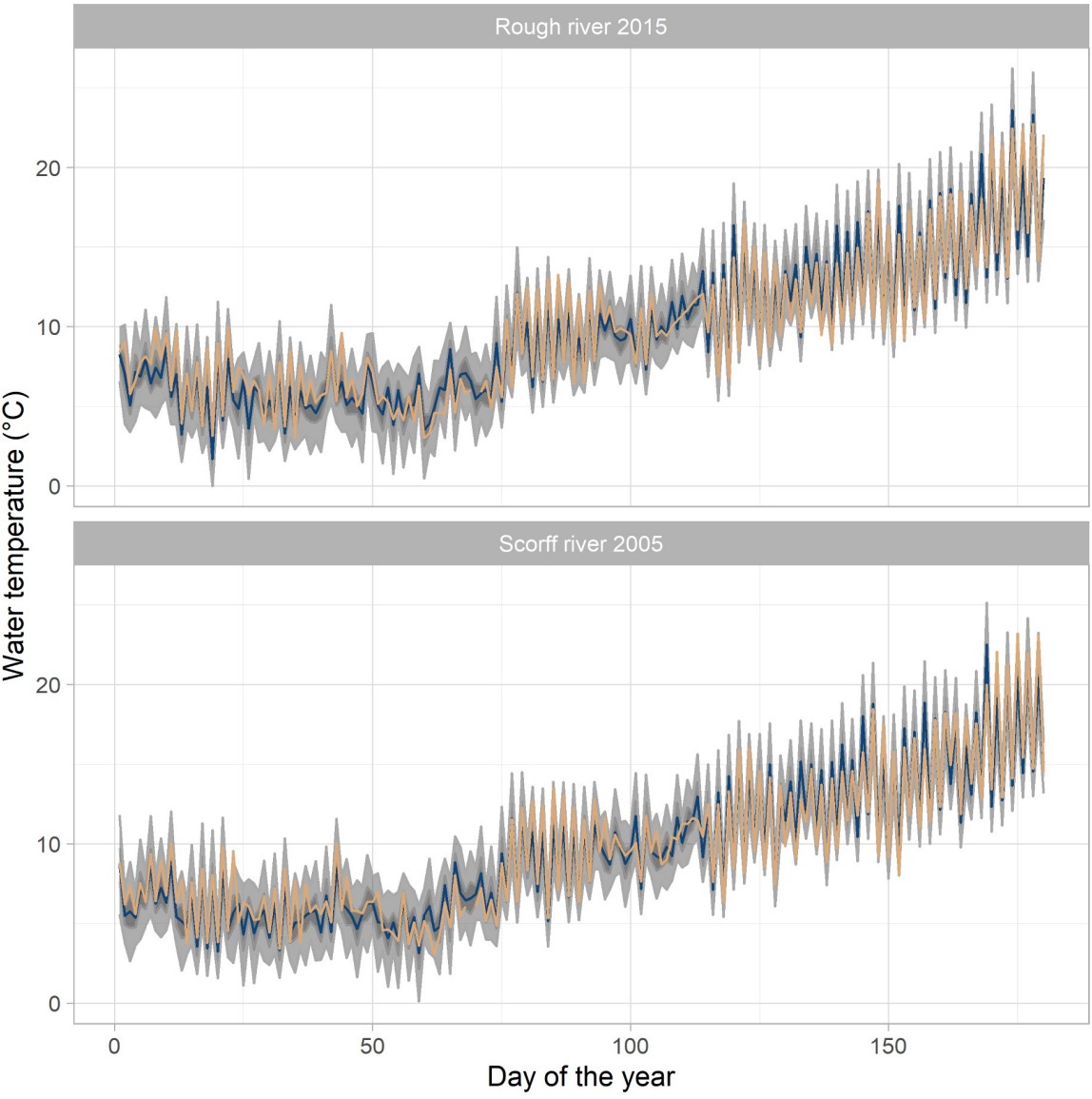

**Fig 4. Observed versus forecasted temperatures coming from the cross validation.** Yellow lines correspond to observed WT temperatures. Blue lines are the median value of predictive posteriors. Grey areas indicate 95% and 50% intervals of the forecasts.

## Posteriors of parameters linking water temperature to the covariates

Modeling the link between the residuals of the different time series resulted in significant differences in $\theta$ posterior distributions (Fig 5). Indeed, three main types of changes were noticeable. First, posteriors of $\theta$ parameters tightened. This was particularly significant on the Scorff River with, for instance, reduction of about 50% in the 95% credible of parameters $\theta_{1'}$, $\theta_2$ and $\theta_{2'}$. Second, posteriors of $\theta_1$ and $\theta_{1'}$ shifted toward stronger positive values. With model $M_2$, the credible intervals of parameters linked to LFL were well centered on 0 compared to that of model $M_1$. Meanwhile, the uncertainty around the regression appears greater with model $M_2$ than with model $M_1$ (see $\sigma$ posteriors, Fig 5).

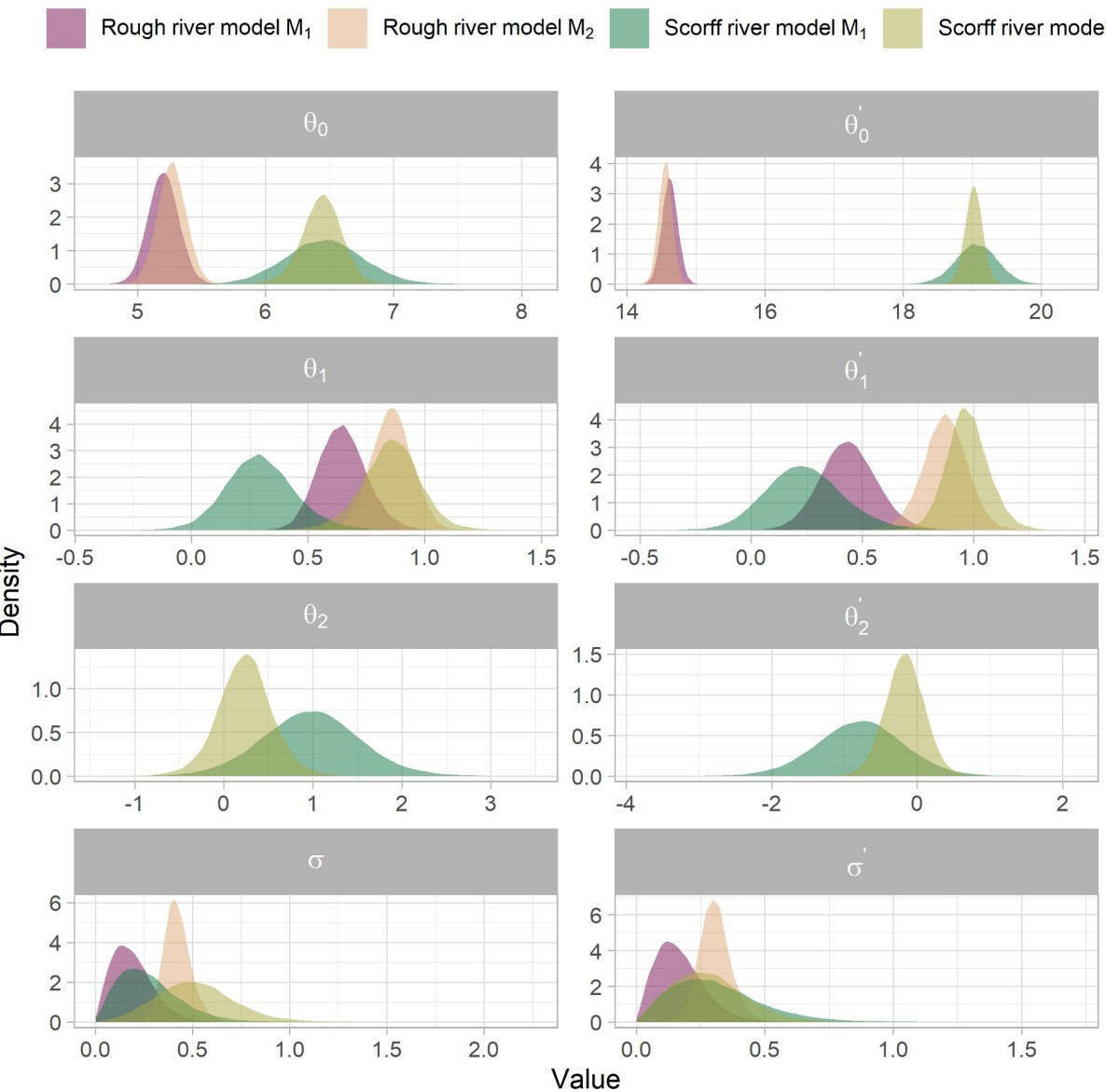

**Fig 5. Comparison of the posteriors of $\theta$ parameters from equation set (2) updated with both model $M_1$ and $M_2$.** The left-hand side of the figure are for parameters linking $max_y^{WT}$ to $max_y^{AT}$ and optionaly $min_y^Q$. The left-hand side of the figure are for parameters linking $min_y^{WT}$ to $min_y^{AT}$ and optionaly $max_y^Q$.

Parameters $\delta$ and $\gamma$ linking daily variations in water temperature to that of the covariates were estimated as positive together with tight posterior distributions on both rivers (Fig 6). All posteriors were centered on values between 0.5 and 0.7 suggesting changes in water temperature to be buffered when compared to those of covariates. The posterior of parameter $\gamma$ appeared about 10 times wider than that of $\delta$ for the Scorff river (0.28 vs 0.03 95% credible interval), suggesting less precision into the impact of flow.

### Forecast warming

Model $M_2$ predicted higher warming of WT than model $M_1$ on both rivers (Fig 7). These differences are directly attributable to the differences in $\theta$ posteriors detailed above (Fig 5). Consequently, the difference between models was greater on the Scorff river. The WT median warming predicted by model $M_2$ for the Scorff closely followed that of AT to peak at 4.59°C

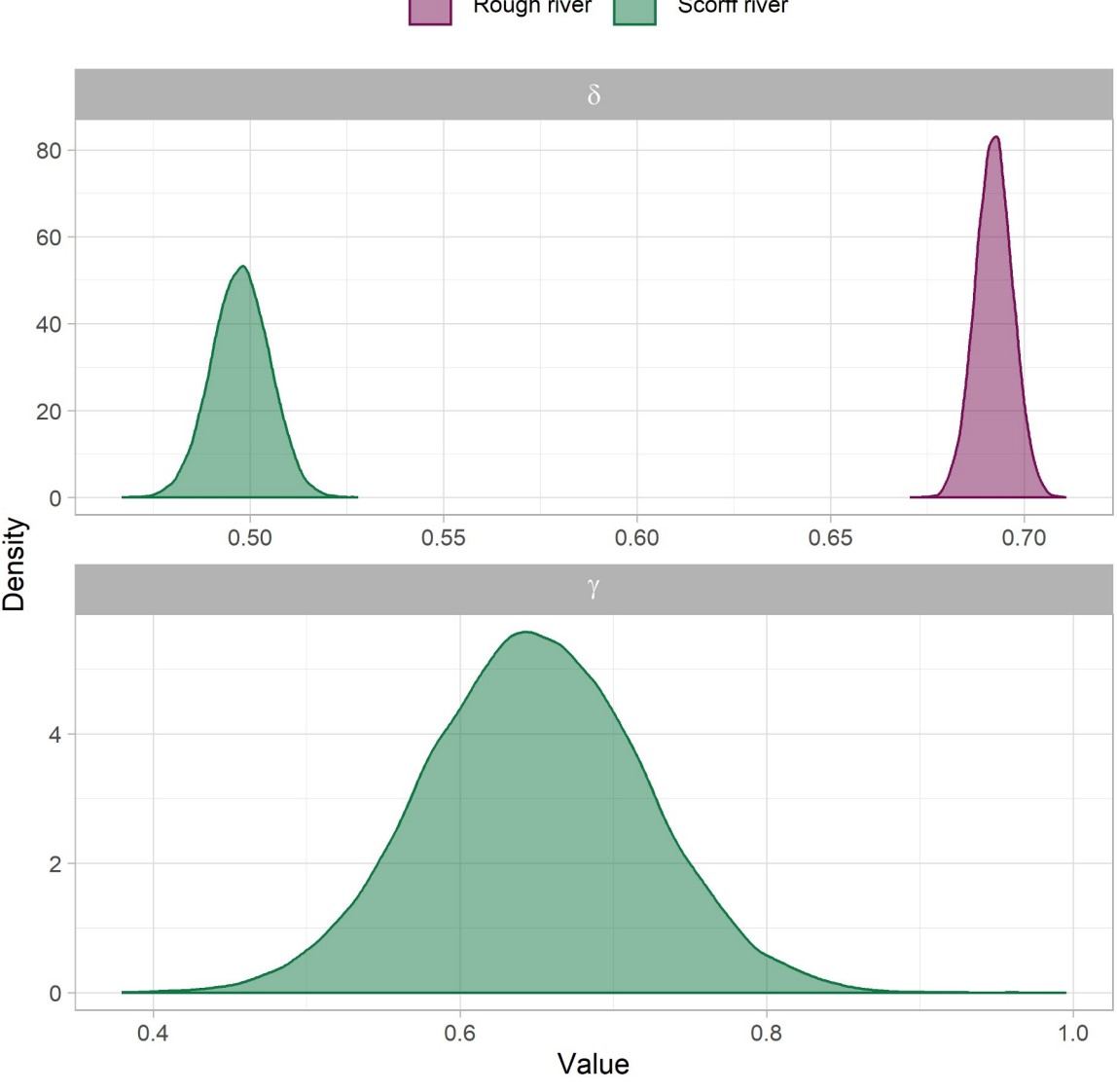

**Fig 6. Posteriors of model M$_2$ parameters linked to daily variations around the seasonal component.** Details can be found in equation set (3).

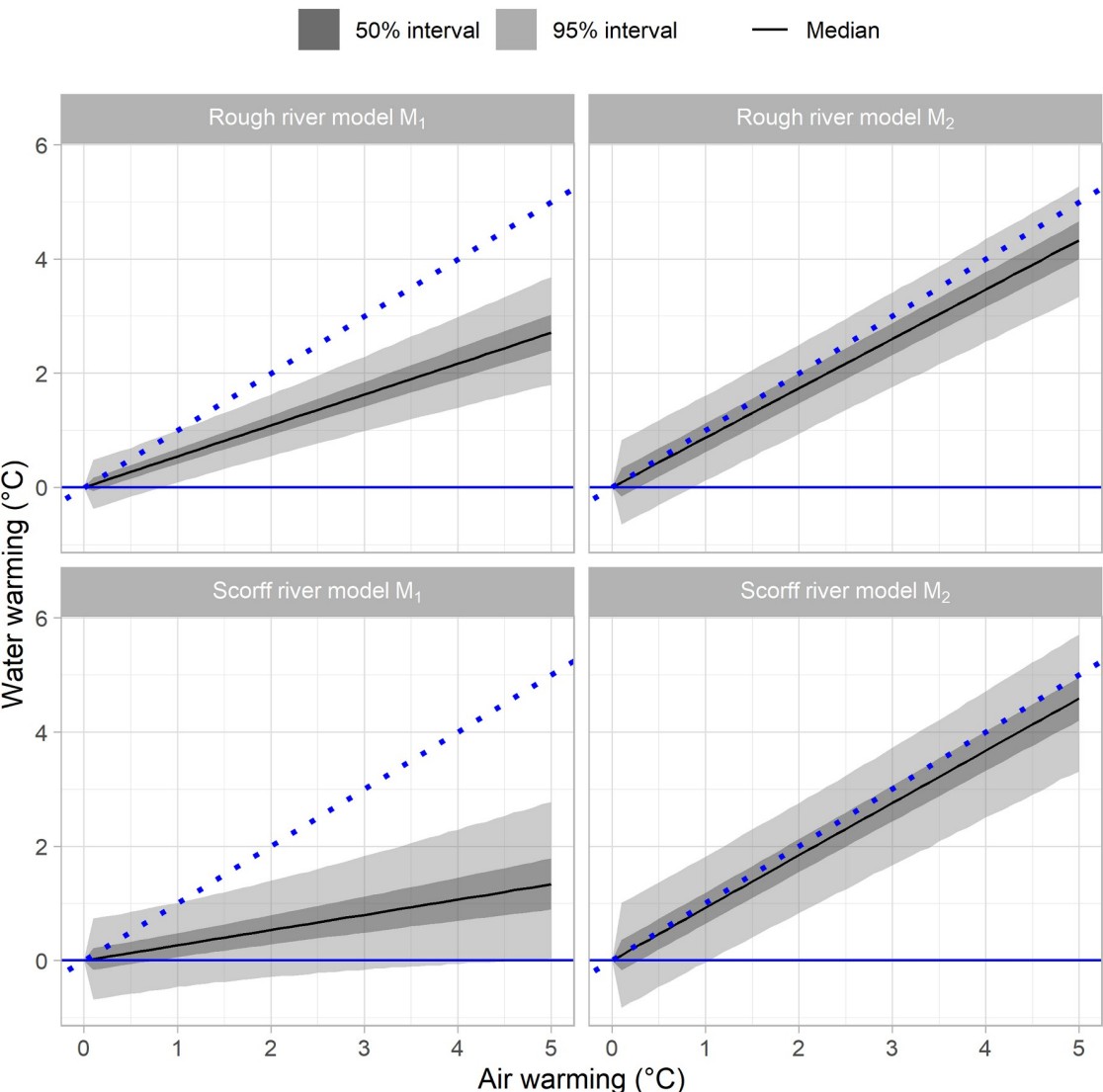

**Fig 7. Warming of stream water temperature predicted across a range of air temperature warming.** The dotted blue line is the 1:1 line.

for an AT warming of 5˚C (median value), indicating that AT warming is still likely to be a bit faster than WT warming there. Meanwhile, model $M_1$ median warming prediction was only 1.33 and its 95% credible interval barely overlapped that of model $M_2$. For the Rough river, the WT warming predictions overlapped more between models but model $M_2$ suggested it could likely be faster there too. Lastly, WT warming predictions coming from model $M_2$ were more uncertain than with model $M_1$. The average 95% credible of predicted WT warming were of 1.88 and 2.01˚C on the Scorff river with model $M_1$ and model $M_2$ respectively for a 5˚C AT warming. On the Rough river, corresponding values were of 1.23 and 1.61˚C.

## Discussion

Expanding the time series decomposition-based approach of [21] to better model daily variation in WT in addition to seasonal variations lead to greatly improved WT predictive abilities.

In doing so, RMSE decreased by about 41% on the Rough river and about 35% on the Scorff River. Discrepancies between 6-month averages of replicated data and observed water temperature were also lower with the new $M_2$ model than with the previous $M_1$ model. This is indicative of a better fit as replicated data generated under model $M_2$ are more similar to observed data. It also means forecast or hindcast 6-month means should be more reliable. The use of an AR1 structure to model WT residuals as in model $M_1$ is very flexible and allows for capturing variability in fit. However, this can be considered as an over-fit when considering our goals. Indeed, model $M_1$ was less successful than $M_2$ at producing long term hindcasts or predictions.

Interestingly, the new model suggested that daily variation in flow may be more important to WT than seasonal variation. Indeed, in case of our new model, the $\theta$ parameters linking the seasonal component of WT to that of flow on the Scorff river appeared centered on 0, meaning that they may not be statistically significant. This appeared surprising as AT and Q six months minima and maxima time series were almost identical between models (see S1 File). In addition, the correlations between Q and AT time series entering the set of Eq (2) remained rather weak, suggesting that the signals were contrasted between covariates. Correlation were even slightly decreasing when going from model $M_1$ to $M_2$, changing from 0.48 to 0.4 for $min_{AT}$ and $max_Q$, and from -0.39 to -0.31 for $max_{AT}$ and $min_Q$. It would be interesting to see if this behavior is also observed on other rivers. Still, the impact of daily flow variations on WT has practical consequences. It indicates that daily management of flow to avoid extremely low and high flow might be useful in the context of climate change adaptation [47]. In particular, ensuring adequate flow in summer will serve to alleviate thermal stress, as flow can mitigate warming. The main issue is that both rainfall predictions by climate models and the subsequent rainfall-runoff models may lack enough precision [48].

The results from model $M_2$ also indicated that WT warming should track that of AT closer than predicted with model $M_1$. In case of the Scorff river, WT potential warming appeared to be at the same scale as that of the AT warming scenario. On the Rough River, the WT forecast appeared less likely to reach 5°C and results suggested that it might be overall slower than on the Scorff river. The more pronounced changes in predictions observed on the Scorff River with our updated model is likely linked to two key points. First it appears that modeling the residuals as done with model $M_2$ allows for a more precise estimate of the long-term component of WT time series than when using model $M_1$. The AR1 structure used in $M_1$ probably absorbed a part of the signal in 6-month values. The Scorff river WT time series were also relatively short and prone to a lot of missing values. Those two things together probably led to an even greater parameter confusion while fitting the signal decomposition. Thus, we observed even more uncertainty in the 6 months min and max estimates from model $M_1$ on the Scorff river compared to those derived from model $M_2$. Lastly, a probable more important WT warming on the Scorff river is also congruent with the energy budget theory [31]. The Scorff river is much longer than the Rough river (75 and 12.3 km respectively). As such, thermal exchanges between air and water are more important and some homogenization is to be expected. Headwaters may thus be way less impacted by climate change as observed on the Rough river and in the USA [49] and their role in mitigating the global warming impact could be important. We note, however, that the opposite observation about the response of headwaters to warming has also been made [50] and warming predictions still overlap significantly in our cases. Further analyses on a range of rivers is necessary to better understand the role of different types of rivers in mitigating climate change.

The overall uncertainty associated with WT warming is high. This will have to be accounted for in decision making. Indeed, whatever the model, the 95% credible interval in WT warming was about 2°C around a given temperature. IPCC scenarios highlight uncertainties in AT

warming in 2100 close to 3.5˚C [46]. The combined uncertainty is very high even over the medium term. Integrating it in a probabilistic framework is important for future ecological studies and decisions making [32]. The approach we developed makes this possible.

Further extensions of our model are possible. For now, the model is limited to one station per river, but may be extended to several. This has already been done for instance with models based on linear [50] and nonlinear regressions [51]. The application could also encompass several rivers as done for instance with regression or neural network [52]. Such an approach could include a hierarchical structure to test for an increased link between AT and WT as the stations are further downstream. The number and length of the time series would have to be chosen carefully as Bayesian fit can be slow when dealing with a large amount of data. Other covariates may also be included in further research. Shading by riparian vegetation can significantly alter WT and vegetation may change due to climate or land use management decisions [24, 53]. The shading factor could be included within the modelling of residuals as reducing extreme values through an additional exponential multiplier.

We offer stream ecologists and stakeholders a generic, parsimonious and effective way to complete, hindcast and predict WT time series in a temperate context. Using only AT and optionally Q time series, it predicts daily WT. The Bayesian framework allows for a full propagation of uncertainties while doing so. A probabilistic rational can then be used for management decision. The code is available on https://github.com/GuillaumeBal/2023.bayes.stream.temperature so readers can perform their own fits. Providing correctly formatted data (locally observed WT, AT and optionally Q data) and the specification a few model options is all that is required to run both models $M_1$ and $M_2$.

## Supporting information

**S1 File. Additional details about models' priors, application time series and models' fits.** (DOCX)

## Acknowledgments

We are grateful to the staff of the Marine Institute, of the 'Unite Expérimentale d'Ecologie et d'Ecotoxicologie Aquatique', of Hydroportail and of Météo France for collecting and providing the data used in this study.

## Author Contributions

**Conceptualization:** Guillaume Bal, Elvira de Eyto.

**Data curation:** Guillaume Bal, Elvira de Eyto.

**Formal analysis:** Guillaume Bal.

**Methodology:** Guillaume Bal.

**Writing – original draft:** Guillaume Bal, Elvira de Eyto.

**Writing – review & editing:** Guillaume Bal, Elvira de Eyto.

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
