## [Decision Letter · Decision Letter 0]

18 May 2023

PONE-D-23-03522Simple bayesian reconstruction and forecasting of stream water temperature for ecologistsPLOS ONE

Dear Dr. BAl

Thank you for submitting your manuscript to PLOS ONE. After careful consideration, we feel that it has merit but does not fully meet PLOS ONE’s publication criteria as it currently stands. Therefore, we invite you to submit a revised version of the manuscript that addresses the points raised during the review process.

We look forward to receiving your revised manuscript.

Kind regards,

Dharmendra Kumar Meena

Academic Editor

PLOS ONE

Journal Requirements:

3. Please ensure that you refer to Figures 8, 9 and 10 in your text as, if accepted, production will need this reference to link the reader to the figure.

4. We note you have included a table to which you do not refer in the text of your manuscript. Please ensure that you refer to Table 3 in your text; if accepted, production will need this reference to link the reader to the Table.

5. We note that Figure 1 in your submission contain map images which may be copyrighted. All PLOS content is published under the Creative Commons Attribution License (CC BY 4.0), which means that the manuscript, images, and Supporting Information files will be freely available online, and any third party is permitted to access, download, copy, distribute, and use these materials in any way, even commercially, with proper attribution. For these reasons, we cannot publish previously copyrighted maps or satellite images created using proprietary data, such as Google software (Google Maps, Street View, and Earth). For more information, see our copyright guidelines: http://journals.plos.org/plosone/s/licenses-and-copyright.

Additional Editor Comments :

The article needs substantial revision in language in terms of grammatical and phrases related mistakes.

Reviewers' comments:

Reviewer's Responses to Questions

**Comments to the Author**

1. Is the manuscript technically sound, and do the data support the conclusions?

Reviewer #1: Yes

Reviewer #2: Yes

2. Has the statistical analysis been performed appropriately and rigorously? 

Reviewer #1: Yes

Reviewer #2: Yes

3. Have the authors made all data underlying the findings in their manuscript fully available?

Reviewer #1: Yes

Reviewer #2: No

4. Is the manuscript presented in an intelligible fashion and written in standard English?

Reviewer #1: Yes

Reviewer #2: Yes

5. Review Comments to the Author

Reviewer #1: Review comments

The work describes the potential application of a conceptualized model for forecasting the global warming. The text describes the impact of global warming on ecological communities and ecosystems worldwide, with a focus on freshwater species. Several works have been reported in this line, and the present work can substantiate to existing knowledge. The overall content is interesting and publishable as it lies within the scope of the journal. At this point, I am against publication of the work, however, suggestions for its improvement are appended.

Abstract

Overall, this is a well-written and informative piece. You provide a clear and concise summary of the four practical steps for mitigating the impacts of global warming on wildlife and highlight the challenges associated with obtaining good temperature data for stream ecologists and managers. You then explain how your expanded hierarchical Bayesian model can help address this data gap by providing daily time series of water temperature and improving our understanding of the uncertainties in warming trends.

Writing style is clear and easy to follow, and you provide specific examples and details to illustrate your points. Additionally, you include technical terms and concepts, but you explain them in a way that is accessible to a general audience.

One suggestion for improvement would be to include more specific details about the case study rivers you used to test the model. For example, you could provide more information about the location of the rivers, their size and characteristics, and any relevant environmental or management factors that might influence their water temperatures. This would help readers better understand the context and relevance of your findings.

Due to the high susceptibility of these species to temperature changes and the limited availability of water temperature time series data, there is a need for models to complete and forecast water temperature. The limitations of existing models are discussed, and a Bayesian hierarchical approach is proposed as a solution. The extended model presented in the text, which includes daily temperature and modeling of daily deviations, is compared to the previous model on two rivers with different conditions. The text provides valuable information and is well-structured, with clear explanations of the different types of models and their advantages and disadvantages.

Discussion

This section discusses the improvements made to a model of water temperature (WT) forecasting using a time series decomposition-based approach. The new model, referred to as M2, takes into account daily variation in addition to seasonal variations, resulting in better WT forecasting abilities. The RMSE decreased by about 30% on the Rough River and 40% on the Scorff River, indicating a better fit. The use of an AR1 structure to model WT residuals in the previous model (M1) was found to be overfitting, resulting in poorer long-term hindcasts or projected forecasts.

The new model also suggests that daily variation in flow may be more important to WT than seasonal variation. The impact of daily flow variations on WT has practical consequences, such as daily management of flow to avoid extremely low and high flow, which might be useful in the context of climate change adaptation. The model also highlights that WT warming should track that of air temperature (AT) more closely than predicted in the previous model.

However, the overall uncertainty associated with WT warming is high, and this must be accounted for in decision making. The approach developed in this study makes it possible to integrate this uncertainty into a probabilistic framework. Further extensions of the model are possible, and it is suggested that the model could be applied to other rivers to determine if the behavior observed is consistent.

My overall view for this submission is revise with minor modifications.

Reviewer #2: A past time series data are use to gives the a reliable result of global warming and mitigating the impacts of globle warming on wild life which effect the maturity size of the species also. As the know global warming is the hot and big issue for the environment as well as population.

6. PLOS authors have the option to publish the peer review history of their article (what does this mean?). If published, this will include your full peer review and any attached files.

Reviewer #1: No

Reviewer #2: **Yes: **Dr. Veerendra Singh

---

## [Author Response · Author response to Decision Letter 0]

19 Aug 2023

A file with our detailedanswers has been provided

---

## [Editor Report · Decision Letter 1]

25 Aug 2023

Simple Bayesian reconstruction and forecasting of stream water temperature for ecologists

PONE-D-23-03522R1

Dear Dr. Bal

We’re pleased to inform you that your manuscript has been judged scientifically suitable for publication and will be formally accepted for publication once it meets all outstanding technical requirements.

Kind regards,

Dharmendra Kumar Meena

Academic Editor

PLOS ONE

Additional Editor Comments (optional):

Article can be accepted for publication.
---

## [Editor Report · Acceptance letter]

7 Sep 2023

PONE-D-23-03522R1 

Simple Bayesian reconstruction and forecasting of stream water temperature for ecologists
A tool using air temperature, optionally flow, in a time series decomposition approach 

Dear Dr. Bal:

I'm pleased to inform you that your manuscript has been deemed suitable for publication in PLOS ONE. Congratulations! Your manuscript is now with our production department. 

Kind regards, 

on behalf of

Dr. Dharmendra Kumar Meena 

Academic Editor

PLOS ONE